# Investigating the Biological Efficacy of Albumin-Enriched Platelet-Rich Fibrin (Alb-PRF): A Study on Cytokine Dynamics and Osteoblast Behavior

**DOI:** 10.3390/ijms252111531

**Published:** 2024-10-27

**Authors:** Emanuelle Stellet Lourenço, Neilane Rodrigues Santiago Rocha, Renata de Lima Barbosa, Rafael Coutinho Mello-Machado, Victor Hugo de Souza Lima, Paulo Emilio Correa Leite, Mariana Rodrigues Pereira, Priscila Ladeira Casado, Tomoyuki Kawase, Carlos Fernando Mourão, Gutemberg Gomes Alves

**Affiliations:** 1Clinical Research Unit, Antonio Pedro Hospital, Fluminense Federal University, Niteroi 24033-900, Brazil; emanuelle_stellet@yahoo.com.br (E.S.L.); neilane_rocha@id.uff.br (N.R.S.R.); renatalb@id.uff.br (R.d.L.B.); rafaelcoutinhodemello@yahoo.com.br (R.C.M.-M.); vhsouzalima@id.uff.br (V.H.d.S.L.); gutemberg_alves@id.uff.br (G.G.A.); 2Graduate Program in Science and Biotechnology, Fluminense Federal University, Niteroi 24210-201, Brazil; leitepec@gmail.com (P.E.C.L.); marianapereira@id.uff.br (M.R.P.); 3Department of Implant Dentistry, Dental School, Fluminense Federal University, Niteroi 24210-201, Brazil; priscilacasado@id.uff.br; 4Division of Oral Bioengineering, Graduate School of Medical and Dental Sciences, Niigata University, Niigata 951-8514, Japan; 5Department of Basic and Clinical Translational Sciences, Tufts University School of Dental Medicine, Boston, MA 02111, USA; carlos.mourao@tufts.edu

**Keywords:** growth factors, bone, L-PRF, PRF, osteoblasts, albumin

## Abstract

The development of effective biomaterials for tissue regeneration has led to the exploration of blood derivatives such as leucocyte- and platelet-rich fibrin (L-PRF). A novel variant, Albumin-Enriched Platelet-Rich Fibrin (Alb-PRF), has been introduced to improve structural stability and bioactivity, making it a promising candidate for bone regeneration. This study aimed to evaluate Alb-PRF’s capacity for cytokine and growth factor release, along with its effects on the proliferation, differentiation, and mineralization of human osteoblasts in vitro. Alb-PRF membranes were analyzed using histological, scanning electron microscopy, and fluorescence microscopy techniques. Cytokine and growth factor release was quantified over seven days, and osteoinductive potential was evaluated with MG-63 osteoblast-like cells. Structural analysis showed Alb-PRF as a biphasic, highly cellularized material that releases lower levels of inflammatory cytokines and higher concentrations of platelet-derived growth factor (PDGF) and vascular endothelial growth factor (VEGF) compared to L-PRF. Alb-PRF exhibited higher early alkaline phosphatase activity and in vitro mineralization (*p* < 0.05) and significantly increased the OPG/RANKL mRNA ratio (*p* < 0.05). These results indicate that Alb-PRF has promising potential as a scaffold for bone repair, warranting further in vivo and clinical assessments to confirm its suitability for clinical applications.

## 1. Introduction

The development and enhancement of technologies applied to tissue regeneration have posed significant challenges in recent decades, particularly with the increasing use of blood derivatives as autologous biomaterials [1]. Due to their bioactivity potential [2], platelet concentrates have been widely utilized in various medical procedures, including the treatment of chronic wounds, osteonecrosis, infrabony defects, plastic and orthopedic surgeries, and as hemostatic agents [2,3].

Initially described by Joseph Choukroun [4], the second generation of platelet aggregates is represented by platelet-rich fibrin (PRF). This biomaterial is obtained through a simplified, low-cost, and completely autologous protocol, involving the centrifugation of peripheral blood aliquots collected in glass or silica-coated tubes without the addition of anticoagulants [5]. The resulting material forms a three-dimensional fibrin network that entraps cells such as platelets and leukocytes, along with growth factors and cytokines, promoting their continuous release over time [2,3,6].

An important issue considered in the application of these materials is the structural stability of the fibrin mesh [7]. Although various protocols for autologous platelet concentrates are tailored for specific clinical applications, limitations in fibrin mesh stability can compromise their effectiveness in certain surgical procedures [8]. For example, the efficacy of PRF as a protective barrier to prevent soft tissue infiltration into bone grafts during guided bone regeneration is not well established [9], even though studies had demonstrated the stability of PRF and its ability to release cytokines for up to 28 days in an in vitro biological environment [10]. Nevertheless, there is no precise estimate of how long this membrane remains intact under a bone graft post-surgery, nor of the effects of enzymes such as metalloproteinases on its degradation, which could impact its efficacy as an autologous barrier for guided bone regeneration [11].

To address this limitation, a protocol for producing albumin-enriched platelet-rich fibrin (Alb-PRF) was proposed by Mourão et al. [12], which has since been the focus of several studies [13,14]. This protocol, based on the denaturation of proteins in the PPP (Poor Platelet Plasma) portion, creates a completely autologous, bioactive biomaterial with superior structural stability compared to conventional PRF, as demonstrated by in vivo studies [15]. Despite its denser ultrastructure, Alb-PRF retains the ability to release cytokines and growth factors typical of blood concentrates. Additionally, Fujioka-Kobayashi et al.’s in vitro study [16] demonstrated that exposure to Alb-PRF stimulated the migration, proliferation, and expression of TGF-β and type I collagen in gingival fibroblasts in vitro, highlighting Alb-PRF as a promising material for regenerative treatments in soft tissues.

Regarding the application of blood concentrates in mineralized tissues, the literature shows positive effects on the proliferation, differentiation, and mineralization of bone cells using various protocols [17]. A clinical study comparing tissue responses to Alb-PRF and PRF implantation in alveoli after third molar extraction observed a reduction in pain, edema, and the release of pro-inflammatory mediators at surgical sites, with more pronounced effects in areas where Alb-PRF was applied [18]. Despite promising clinical evidence, there is still a lack of detailed knowledge regarding the dynamics of bioactive molecule release by Alb-PRF and its impact on the cellular functions of bone cells, as compared to widely employed protocols such as L-PRF. In this context, the present study aims to evaluate Alb-PRF in terms of its structural characteristics and capacity for active cytokine and growth factor release. Additionally, the study assessed the ability of Alb-PRF to stimulate in vitro proliferation, differentiation, and mineralization of human osteoblasts, which are key markers of the bone regeneration process mediated by these biomaterials.

## 2. Results

The structures of the Alb-PRF and L-PRF membranes are evidenced in the histological sections shown in Figure 1. The fibrin network, stained red, can be identified along both membranes (Figure 1A,B). In the L-PRF membrane (Figure 1A), it is possible to identify a loosely distributed fibrin web, with the buffy coat area being visualized at its end, containing a high density of nucleated cells. The biphasic nature of Alb-PRF is confirmed by the presence of a dense denatured albumin mass (Figure 1B), where sparse nucleated cells can also be observed.

Figure 2 shows the ultrastructure of the Alb-PRF and L-PRF membranes from micrographs obtained by scanning electron microscopy (SEM), after 1, 14 and 28 days of incubation in a culture medium. The L-PRF membrane is characterized by a fibrin mesh with closely entangled fibers, where cells are entrapped. The Alb-PRF membrane presents again a dense surface with an evident layer of denatured protein deposition, completely covered with fibrin fibers, where it is possible to observe the entrapment of cells and platelets. Differences in the pattern of fiber organization can be observed on the first day. The L-PRF membrane is characterized by the fibrin network inherent to PRF and evident blood cells, while the Alb-PRF membrane forms a mass covered with proteins. Even after 2 weeks, the two membranes appear intact compared to the first day after production. On the 14th day, the mass (barrier) imposed by Alb-PRF can be observed once again, in contrast to the L-PRF fibrillar framework, making it possible to detect the presence of cells. On the 28th day, it is observed that the general pattern does not change; however, in L-PRF, the membrane presents large grooves, suggesting a change in the structure of the membrane, even though the appearance of the fibrin network appears intact, and with the presence of cells. However, there are no major observable changes in the Alb-PRF membrane.

Figure 3 demonstrates the distribution of cells within the fibrillar structures of the L-PRF and Alb-PRF membranes. Nuclei labeling revealed the presence of nucleated cells in all areas analyzed, with increasing amounts starting from the upper portion of the L-PRF membrane, showing a high cell density in the lower portion (buffy coat). The Alb-PRF membrane, on the other hand, presents a relevant cellular distribution across its entire length.

The curve referring to the quantification of total protein elution from the membranes is represented in Figure 4. It is observed that the concentration of total proteins present in the conditioned medium of the L-PRF sample is higher than the other membranes on day 1. However, the L-PRF and Alb-PRF show a very similar release pattern throughout the remaining experimental times. By applying a linear regression, it was possible to determine a similar pattern of protein release rate for L-PRF (0.07742 mg/day) and Alb-PRF (0.07018 mg/day).

Figure 5 shows the quantification of 27 analytes investigated in culture medium samples exposed to the three different membranes over periods of 1 and 7 days. Most of the molecules studied were detected on the first day of elution, including a range of pro- and anti-inflammatory cytokines, as well as high concentrations of growth factors, the most notable being VEGF and PDGF-BB, with emphasis on the high levels of this growth factor in Alb-PRF on day 1 and in the L-PRF membrane on day 7 of elution. On the seventh day of elution, it is possible to observe a pattern of intensified release of some chemokines, cytokines, and PDGF-BB, as well as a reduction in other elements related to inflammatory dynamics. A slight reduction in VEGF concentrations and a large release of IL-6 in the L-PRF membranes were also observed.

The cell viability of osteoblasts exposed to different concentrations of extracts from L-PRF and Alb-PRF membranes is shown in Figure 6. High biocompatibility values are observed in all concentrations of Alb-PRF, while L-PRF was cytotoxic at concentrations higher than 25% of dilution. Cell viability above 100% of the control at lower dilutions indicates a possibility of induction of proliferation already at 24 h by both biomaterials.

After exposing osteoblasts to 25% extracts of the different materials for periods of 1 to 7 days, the induction of proliferation can be observed by a direct correlation between the increase in cell density and time (Figure 7). An increase in cell density was observed in all samples exposed to conditioned media, in a similar way on days 1 and 3. On day 7 there was no significant difference between the experimental groups, unlike the comparison with the control group, which showed a significant difference, with *p* < 0.05).

Figure 8 shows the effects of exposure to the extracts of L-PRF and Alb-PRF on alkaline phosphatase (ALP) activity secreted by osteoblasts in the culture media, measured over multiple time points (1, 3, 7, 14, and 21 days). The activities of this enzyme were significantly different between all groups at days 3, 7, 14, and 21 (*p* < 0.05). Osteoblasts exposed to Alb-PRF exhibited a large peak of ALP activity on the first day, while the L-PRF and positive control (mineralization medium) demonstrated a more gradual increase, with the highest ALP activity detected at day 21. Notably, Alb-PRF exhibited significantly higher ALP activity than L-PRF and the control at most time points (days 1 to 14 (*p* < 0.05)). However, by day 21 the cells exposed to L-PRF produced levels of ALP activity similar to Alb-PRF (*p* > 0.05), both being higher than the positive mineralization control (C+).

Figure 9 shows the quantification of alizarin red after exposure of osteoblasts to 25% conditioned media. The data indicate a time-dependent increase in mineralization across all experimental groups. While there was no difference between groups on the first day of treatment, by day 7, all treated groups, including the positive control (C+), Alb-PRF, and L-PRF, induced higher levels of mineral deposition that the non-treated control. Notably, Alb-PRF demonstrated significantly higher mineralization compared to the other groups at each time point, achieving almost twice the staining intensity observed in the negative control group (*p* < 0.05).

Figure 10 presents the evaluation of the expression levels for various inflammatory and bone metabolism-related markers in the osteoblasts after 1 day of exposure to 25% conditioned media from the tested biomaterials. The results indicate that there were no statistically significant differences (*p* > 0.05) in the expression of the inflammatory mediators (IL-6, IL-8, IFN-y) and the anti-inflammatory marker IL-10 between the groups (Figure 10A–D). This suggests that the treatments did not significantly alter the inflammatory or anti-inflammatory responses in the MG-63 cells under the conditions tested. For bone metabolism markers, the Alb-PRF group showed significant differences compared to both the control and L-PRF groups across all relevant panels. Both OPG and RANKL were significantly upregulated in the Alb-PRF group (Figure 10F,G), while these differences resulted in a significantly higher OPG/RANKL ratio (1.4) as compared to the control (0.27) and L-PRF (0.80), which was also significantly higher than the control (*p* < 0.05). Additionally, panel 10E shows that FGF2 expression was significantly higher in the Alb-PRF group compared to both the control and L-PRF groups (*p* < 0.05).

## 3. Discussion

Despite providing favorable characteristics for tissue regeneration, the residence time of the PRF membrane in vivo makes its application as a suitable scaffold for bone regeneration questionable [9]. Faced with such a limitation, our research group proposed a new platelet aggregate protocol, the fibrin membrane enriched with denatured albumin (Alb-CGF) which, in addition to providing the material with the biological characteristics originating from a platelet concentrate, also presents a slow degradation of its structure over time [12,15]. This improvement is achieved through the inclusion of denatured albumin, obtained by thermal processing of the PPP fraction (Poor Platelet Plasma), with the fibrin protein content [13]. In this work, we show that this rationale may also be applied to a PRF protocol employing horizontal centrifugation, a modification previously proposed for a more homogeneous cellular distribution and release profile of growth factors and cytokines [10,19].

The characterization of the resulting membranes (Alb-PRF) by SEM and histological analysis revealed a malleable biphasic material, with a dense fibrin network with characteristics very similar to classic PRF, such as the fibrillar appearance and the entrapment of different cell types [20,21], surrounded by a dense denatured albumin mass, all typical features described in the literature for this material [12,15]. It includes the detection of nucleated cells all along the Alb-PRF membranes, confirming the ability to trap nucleated cells. Histological sections and fluorescence microscopy show these cells interspersed in the denser fibrillar structure of the material, with a homogeneous distribution throughout the different (upper, middle and lower) regions. It is hypothesized that such cellular configuration can contribute to a more homogeneous supply of bioactive molecules throughout the blood concentrate, unlike the cellular polarization found in L-PRF [4,20]. This characteristic could possibly provide a more uniform biological action, enabling the distribution of growth factors and cytokines throughout the entire contact area of the biomaterial with the tissue to be regenerated.

A potential drawback of this protocol could be a possible excessive release of albumin, which could lead to inflammation, fibrosis, altered mechanical properties, disruption of local homeostasis, and edema [22]. Nevertheless, the results show that, at least in vitro, Alb-PRF and the already clinically tested L-PRF release very similar levels of detectable proteins in biological media, despite their supposed high protein density, given the addition of denatured albumin concentrate during their preparation. These results corroborate the in vitro findings of other studies, where there was preservation of the three-dimensional structure and prolonged release of analytes [10,15].

The impact of growth factors released from fibrin membranes on tissue regeneration processes is recognized as one of the main features of L-PRF, whether through the promotion of angiogenesis, migration, proliferation or differentiation of mesenchymal cells, fibroblasts and osteoblasts [23]. The analysis of the comparative release of bioactive molecules from the membranes in vitro showed high levels of the growth factors VEGF and PDGF-BB in both materials at 1 and 7 days of extraction. On the other hand, a considerably higher release of PDGF-BB was observed from the Alb-PRF membrane on the first day of elution. In addition to its positive action on cell proliferation, PDGF-BB has been described as a molecule that induces both cell proliferation, differentiation, and mineralization of osteoblasts through the activation of the Src/JAK2 [24], as well as an increase in the expression of alkaline phosphatase, an important marker of the commitment with mineralization [17].

Pro- and anti-inflammatory cytokines are released in significant quantities in the Alb-PRF, with the L-PRF membranes presenting increased levels of IL-6, IL-8, G-CSF and IL-1RA. Although the L-PRF membranes can serve as immunomodulatory nodules, excessive expression of inflammatory mediators can negatively affect the tissue repair process, especially in bone, where molecules such as IL-6 can act in osteoclastogenesis, impacting the balance between new bone formation and absorption. In this sense, the Alb-PRF membranes have a milder proinflammatory in vitro profile [16], with the gradual release of several cytokines at lower levels than L-PRF. This is compatible with in vivo results showing excellent biocompatibility on a subcutaneous animal implantation model [15].

This is one of the first reports investigating the effects of Alb-PRF with bone cells, using MG-63 osteoblast-like cells originating from a human osteosarcoma. Even though these are aneuploid cells with greater proliferation rates, they are often used as a bone cell model due to their similarities to human normal osteoblasts in terms of mineralization, differentiation, and general behavior, coupled with the advantage of result standardization provided by the use of a well-established cell line [25,26]. Cellular behavior upon exposure to media conditioned with either L-PRF or Alb-PRF revealed interesting effects for tissue regeneration processes.

Cell viability was assessed through mitochondrial/metabolic activity, confirming higher cell densities in the L-PRF and Alb-PRF groups compared to the control group, suggesting induction of cell proliferation [21,27]. However, the L-PRF membrane exhibited significant cytotoxicity at higher concentrations. This phenomenon could be partially attributed to the release of inflammatory cytokines, which are known for their cytotoxic potential. Indeed, our results showed elevated levels of key cytokines, including IL-6, IL-8, TNF, and IFN, which likely contribute to the observed effects [28]. Nevertheless, while the release of inflammatory cytokines may be an important factor, other mechanisms could also explain the cytotoxicity at higher concentrations of L-PRF, such as osmotic stress due to the high concentration of bioactive molecules and ions, potentially disrupting cellular function through an imbalance in the culture medium. Additionally, the exposure to growth factors might overstimulate certain cellular pathways, leading to dysregulated signaling and, in some cases, apoptosis. Changes in the pH of the culture medium, resulting from the breakdown of L-PRF components at higher concentrations, may also contribute to cellular stress and cytotoxicity by disrupting the conditions necessary for optimal cell metabolism. Moreover, the presence of platelet-derived microvesicles (PMVs) in L-PRF, which carry a variety of bioactive molecules, may also provoke unintended cellular stress [29], further contributing to cytotoxic outcomes. Thus, the cytotoxicity observed in the L-PRF group at higher concentrations is likely multifactorial and, while it is reported in several different in vitro studies [17], further assessments should dissect these mechanisms and clarify their relative contributions to the cytotoxic effects seen with high concentrations of L-PRF.

The positive effect of different formulations of platelet aggregates on cell proliferation has been extensively studied [30,31,32], and the impact of these biomaterials on the processes of differentiation and mineralization in osteoblasts has been recently reviewed [2,17]. Using the MG-63 model, it was possible to observe an increase in cell density over time through the presence of extracts, where the L-PRF and Alb-PRF membranes induced greater cell proliferation compared to the control group, showing results compatible with different correlated studies [16,33,34]. A positive effect on cell proliferation, that may accelerate repair processes supporting tissue neoformation, is expected from bone cells exposed to growth factors released by PRF membranes [17].

Given the stimuli from the extracts, it was possible to observe that MG-63 cells were induced to initiate the mineralization process of their extracellular environment, since there was greater deposition of minerals based on calcium phosphate and higher dosages of alkaline phosphatase in all experimental groups, compared to control groups. A peak in alkaline phosphatase was observed on the first day of the experiment in the groups exposed to Alb-PRF extracts, suggesting the acceleration of processes related to mineralization. Alkaline phosphatase (ALP) is an enzyme secreted by osteoblasts in the early stages of differentiation to control the process of bone mineral deposition and is involved in calcification and modification of the matrix so that there is deposition and growth of crystals [35]. The increase in the activity and expression of this enzyme in a culture of osteogenic cells signals the beginning of the processes of mineralization and differentiation. Routinely, the intensification of production of this enzyme can only be detected from the seventh day of induction in control groups exposed to known inducing substances [36,37]. However, the change in the ALP detection pattern at shorter times, induced by the Alb-PRF extract, reaching levels ten times higher than L-PRF, is very characteristic of materials that accelerate the bone repair process. Even though the enzyme activity reaches comparable levels at the 21st day, it happens with at least a week of delay, reinforcing the suggestion that the effects of Alb-PRF are related to an acceleration of the in vitro mineralization process. Furthermore, it has been demonstrated that osteoblasts and preosteoblasts tend to shift the in vitro ALP release peak to shorter experimental times when exposed to growth factors such as PDGF, similarly to the pattern observed for Alb-PRF in this study, followed by an increase in mineralization in the subsequent days [24]. Interestingly, we can observe a strong (significantly greater, *p* < 0.05) release of PDGF by Alb-PRF, as compared to L-PRF, which may be part of the explanation for these differences regarding ALP activity.

Interestingly, when the ability of deposition of calcium minerals by the exposed cells was accessed by the alizarin-red assay, there was evidence of intensification of the mineralization process after exposure to Alb-PRF extracts starting from 7 days of exposure. It is important to point out that the high alkaline phosphatase activity, noticeable on the first day of exposure to the extract, directly correlates with the phenotypic evolution presented by the alizarin dosage over the experimental period of 7 days, which was significantly higher than L-PRF and the mineralization control group (*p* < 0.05) until the longest experimental time investigated (21 days). This interesting result is among the first evidence that Alb-PRF may present some osteoinductive characteristics, supported by a recent in vitro study [34], but still demanding in vivo and clinical confirmation.

Potentially interesting properties of Alb-PRF may also be revealed by the preliminary molecular analysis of mRNA expression performed in this study. The data indicate that, while L-PRF and Alb-PRF were very similar in not promoting any effect on the relative expression of inflammatory cytokines (IL-6, IL-8, INF, IL-10), as compared to the unexposed control, exposure to Alb-PRF was unique in activating other bone-related markers. Cells exposed to Alb-PRF increased their expression of FGF2, which is reported to enhance the integration of biomaterials with surrounding bone tissue, reducing healing time and improving long-term stability of implants. Furthermore, exposure to Alb-PRF altered both the expression of RANKL and OPG mRNA, resulting in an increased expected OPG/RANKL ratio. An increase in this ratio is interpreted as highly advantageous for bone regeneration, as it promotes osteoblast activity while inhibiting osteoclast-mediated bone resorption [38]. OPG (osteoprotegerin) acts as a decoy receptor for RANKL (receptor activator of nuclear factor kappa-Β ligand), preventing RANKL from binding to its receptor on osteoclasts, thereby reducing their formation and activity. Therefore, a shift toward a higher OPG/RANKL ratio may help maintain bone mass, enhance bone formation, and foster a favorable environment for long-term bone stability [39].

Given the lack of knowledge about the possible biological impacts of Alb-PRF, this study takes the first steps towards understanding these events based on exposure to human cells. This approach allowed the investigation of several parameters related to the regenerative potential and biological safety of this novel material. Our study used an indirect exposure model to investigate the biological effects of Alb-PRF on osteoblast behavior, an approach that has been extensively used in PRF-related studies due to its ability to isolate critical parameters such as proliferation, mineralization, and molecular release [40,41,42,43]. While the indirect exposure model minimizes interference from the fibrin matrix, which could entrap bioactive molecules and complicate quantification, future studies involving direct cell membrane interaction will contribute to evaluating additional factors such as cell migration into the L-PRF scaffolds [44,45,46,47] and Alb-PRF’s potential barrier effects and their implications for clinical applications. In addition to these biological insights, future studies should focus on evaluating the structural properties of Alb-PRF, which are crucial for its function as a scaffold in tissue regeneration. Mechanical assessments, such as measuring stiffness, elasticity, and porosity, will provide a better understanding of how Alb-PRF supports cell migration and differentiation. Techniques like atomic force microscopy (AFM), differential scanning calorimetry (DSC), rheometry, and scanning electron microscopy (SEM) will help analyze these properties, complementing our current findings and offering a more comprehensive picture of Alb-PRF’s potential in bone regeneration.

Nevertheless, in vitro systems present the difficulty of mimicking the complexity of certain biological interactions, requiring future in vivo and clinical studies for further clarification. The biocompatibility and structural stability of Alb-PRF were investigated in animal pre-clinical standardized assessments [15], evidencing the significant stability and biocompatibility of Alb-PRF, with maintenance of biomaterial volume in the implantation area and absence of significant inflammatory reaction. A clinical study carried out by Javid and collaborators [18] confirmed the biocompatibility of Alb-PRF in human tissues and evidenced a reduction in post-surgical pain after third molar extraction, with a lower local release of inflammatory cytokines as compared to L-PRF surgical sites. In this context, the present in vitro findings, taken together, should raise interest in performing both pre-clinical and clinical studies investigating this new material in other types of surgical interventions that will allow determining the outcomes of its potential applications in procedures involving bone repair.

## 4. Materials and Methods

### 4.1. Ethical Considerations

Blood samples were collected, upon consent, from nine healthy research participants of both sexes and aged between 30 and 45 years. The study was approved by the Ethics Committee of the Hospital Universitário Antônio Pedro (CAAE 12126919.7.0000.5243) and carried out following the ethical standards of the 1975 Declaration of Helsinki, revised in 2000.

### 4.2. Preparation of L-PRF Membranes

To establish the standard protocol (control group), samples of platelet- and leukocyte-rich fibrin membranes (L-PRF) [4] were obtained from centrifugation of 9 mL tubes containing peripheral blood, without the addition of any substances (Vacutube, Biocon^®^, São Paulo, Brazil). The processing of these samples was carried out using the protocol of 2700 rpm for 12 min (~700 RCF-max) in the DUO fixed-angle rotor centrifuge (DUO^®^Process for PRF, Nice, France). After polymerization of the contents inside the tube, the material was removed with surgical forceps and, with the aid of a piece of sterile gauze, slight compression was applied, producing a gelatinous structure.

### 4.3. Preparation of Alb-PRF Membranes

Peripheral blood was collected from four healthy participants with no history of using anticoagulant medication, using 9 mL glass tubes, without the addition of any substances (Vacutube, Biocon^®^, São Paulo, Brazil). To produce each membrane, two tubes were positioned in a swing-out centrifuge with a horizontal rotor (Bio-PRF, Venice, FL, USA), and the protocol for H-PRF was applied to obtain the liquid phase (plasma + portion rich in cells)—centrifugation for 8 min at 700 RCF-max. This protocol is based on the adaptation of the production parameters of the Alb-CGF membrane [12], through the use of horizontal centrifugation in the Bio-PRF centrifuge (Bio-PRF, Venice, FL, USA). After processing, it was possible to observe the blood plasma and the remaining decanted material containing red blood cells. Two milliliters of the initial portion of plasma (platelet-poor plasma [PPP]) were collected with a syringe (Injex^®^, São Paulo, Brazil), while the other portions of blood (buffy coat, LPCGF, and red blood cells) were reserved at room temperature (20 °C). The syringes containing PPP were inserted into a device to activate plasma albumin gel by denaturation (APAG^®^, Silfradent, St. Sofia, Italy). After 10 min of processing at 75 °C, the syringes were stored at room temperature for another 10 min and protected from light (as recommended by the manufacturer). The denatured albumin was deposited in a glass container to obtain the format of interest. Subsequently, using a 10 mL syringe (Injex ^®^, São Paulo, Brazil) and an 18 G hypodermic needle (Injex ^®^, São Paulo, Brazil), the liquid fraction of the growth factor concentrates and the buffy portion coat were collected in an approximate volume of 4 mL, added to the denatured albumin in the glass container, and then gently mixed with tweezers. After waiting for the fibrin polymerization process (approximately 5 min), the membrane was obtained with the previously established format.

### 4.4. Histological Analysis

After completing the polymerization phase, the membranes were fixed in a 4% paraformaldehyde solution and immersed in 15% and 30% sucralose solutions. Next, they were embedded in OCT and frozen using liquid nitrogen vapor and subsequent storage in a freezer—80 °C. Histological sections were performed in a cryostat (Leica Biosystems, Nussloch, Germany) with a thickness of 7 μm and stained with hematoxylin and eosin (HE). The photos were taken under an optical microscope (Axio A1, Zeiss, Jena, Germany) at 10× and 20× magnification.

### 4.5. Scanning Electron Microscopy

Scanning electron microscopy analysis was carried out after the production of the membranes in order to evaluate their structure. Analyses were carried out on 7, 14, 21, and 28 days to investigate changes in the fibrin network. Each portion was fixed with Karnovsky’s solution and post-fixed with 0.2 M sodium cacodylate and 1% osmium tetroxide solution, finally being dehydrated in alcohol solutions (ranging from 15% to 100%) and hexamethyldisilazine (HMDS). The materials were metalized with gold and observed at 15 kV with a scanning electron microscope (JEOL JSM- 6490 LV, JEOL Ltd., Akishima, Japan).

### 4.6. Fluorescence Microscopy

Fluorescence microscopy analysis was performed in order to evaluate the distribution of nucleated cells along the membranes. Samples were fixed by treatment in 4% paraformaldehyde solution for 15 min, and their cell nuclei were marked by exposure to DAPI (4′,6-diamidino-2-phenylindole) diluted 1:5000 in phosphate buffered saline (PBS). The samples were observed with a 20× objective under an inverted fluorescence microscope (Axio Observer A1, Zeiss, Jena, Germany) and photographed with a digital camera (Axiocam Rev.3 MRc, Zeiss, Jena, Germany). Quantification of nucleated cells was performed using Image-Pro Plus 6.0 software (Media Cybernetics, Rockville, MD, USA).

### 4.7. In Vitro Elution of Cytokines and Growth Factors

The Alb-PRF membranes were cultured for up to three weeks after their manufacture to analyze the dynamics of cytokine and growth factor release over time. For this, membranes were incubated in triplicate in 6-well culture plates (TPP, Trasadingen, Switzerland), in the presence of 4 mL of DMEM medium (Dulbecco’s Modified Eagle’s Medium, GIBCO, Waltham, MA, USA), without the use of antibiotics, in a humid atmosphere at 37 °C and 5% CO_2_. Aliquots of the extracts were collected at 1 and 7 days of culture and stored in a freezer at −80 °C. The extracts were used to measure growth factors/cytokines, and for in vitro bone cell induction assays.

### 4.8. Quantification of Total Proteins

In order to comparatively analyze the release of proteins from the membranes, considering the high content of albumin from the Alb-PRF and its possible clinical effects, total proteins were quantified using the Bradford test, present in the extracts collected at experimental times of 1, 7, 14 and 21 days. The test was performed with the commercial kit (Bradford Reagent, Sigma-Aldrich, St. Louis, MO, USA) measured by spectrophotometry at an optical density of 595 nm (Synergy II—Reader, Biotek, Winooski, VT, USA).

### 4.9. Evaluation of the Release of Cytokines and Growth Factors

The concentrations of cytokines and growth factors released by osteoblasts were measured through the analysis of conditioned media after 1 or 7 days of cultivation in culture media. To detect biomolecules, a multiparametric immunoassay was used, based on magnetic microbeads labeled with XMap technology (LuminexCorp, Austin, TX, USA), using a commercial kit (27-plex panel, Bio-Rad, Hercules, CA, USA) capable of quantifying IL-1β, IL-1RA, IL-2, IL-4, IL-5, IL-6, IL-7, IL-8 IL-9, IL-10, IL-12 (p70), IL-13, IL-15, IL-17, CCL11, FGF-b, CSF3, CSF2, IFN-γ, CXCL10, CCL2, CCL3, CCL-4, PDGF, CCL5, TNFα and VEGF. Quantification of magnetic beads and dosages was performed with a Bio-Plex MAGPIX system (Bio-Rad). Results were analyzed using Xponent v. 3.0 software (Luminex Corp, Austin, TX, USA).

### 4.10. Cell Culture

To carry out the in vitro testing of the effects of biomaterials on bone cells, MG-63 osteosarcoma cells, purchased from the Rio de Janeiro Cell Bank, were used due to their similarities with mature osteoblasts and their recognized capacity for mineralization and differentiation. The cells were grown in 75 cm^2^ culture bottles in DMEM—Dulbecco’s culture medium, Modified Eagle’s Medium—High Glucose (Cultilab, Campinas, Brazil) supplemented with 5% fetal bovine serum (Gibco, Thermo Fisher Scientific, Waltham, MA, USA) and maintained in greenhouses at 37 °C at 5% CO_2_.

### 4.11. Cytocompatibility Assessment

Cells were used close to total confluency and with viability > 95% using trypan blue (0.4% in PBS) and arranged in a 96-well plate, at a density of 2 × 10^4^ cells per well, presenting quintuplicate replicates. After 24 h, the culture medium was replaced by 200 μL of the extract in the wells corresponding to the experimental groups. The original extract was considered as a 100% extract, to which further culture media was added, producing 3 other dilutions: 50%, 25% and 12.5%. As a positive control, 100% latex extract was used and, as a negative control, a culture medium supplemented with 5% FBS was used. The viability of osteoblasts after exposure to the different dilutions of the biomaterial extracts was evaluated using the XTT (2,3-bis-(2-methoxy-4-nitro-5-sulfophenyl)-2H-tetrazolium-5-carboxanilide) test (In Cytotox, Xenometrix, Allschwil, Arlesheim, Switzerland). The XTT test is based on the ability of mitochondrial enzymes to convert soluble tetrazolium salts into soluble formazan compounds, quantified by a spectrophotometer at a wavelength of 480 nm.

### 4.12. Analysis of the Effects of Membrane Extracts on Cell Proliferation

To evaluate the cell proliferation parameter, the cells were exposed to conditioned media for periods of 1, 3, 5, and 7 days, and cell density was evaluated using the CVDE assay (crystal violet dye exclusion test). Commercial kits were used in this test (In Cytotox, Xenometrix, Allschwil, Arlesheim, Switzerland). The crystal violet test (CVDE) is a simple assay that evaluates the density of cells in a well by marking their DNA, after treatment and washing of excess dye, with the absorbance at 540 nm being equivalent to the number of cells in the well.

### 4.13. Analysis of the Effect of Enriched Medium on Mineralization

MG-63 cells were exposed to the described extracts to evaluate the osteoinductive potential of the medium rich in growth factors/cytokines. The assay was performed in triplicate. The cells were plated in 24-well plates at a density of 2 × 10^4^ cells and, when they reached confluence, they were exposed to the extract obtained after 24 h of exposure to the membrane extracts at a concentration of 25%. The conditioned medium with the extracts was changed every 72 h, and the cells were maintained for up to 21 days. Exposure to an unconditioned culture medium represented the negative control, while the positive control involved the exposure to a medium supplemented with osteoinductive conditions (0.2 mM ascorbic acid and 10 mM β-glycerolphosphate). All groups were supplemented with 5% FBS and 1% streptomycin/penicillin. The supernatants of cell cultures after each experimental time were collected and stored in a freezer at −80 °C for subsequent analysis of the activity of the alkaline phosphatase enzyme using a commercial kit (Labtest Diagnóstica, São Paulo, Brazil).

To quantify calcium deposits, alizarin was solubilized by adding 400 μL of 10% acetic acid in ultrapure water, and the bottom was scraped and transferred to one Eppendorf tubes; the tubes were centrifuged at 14,000 RPM for 10 min to collect the supernatant, new tubes were transferred, and their pH was adjusted until it was in the range of 4.1 and 4.5. Quantification of the released dye was performed using a spectrophotometer at 405 nm (Sinergy II, Biotek, Winooski, VT, USA).

### 4.14. Evaluation of Protein Expression

After 1.0 × 10^6^ MG-63 cells were exposed to the extracts for 24 h inside cell culture flasks, cells were scraped and total RNA was extracted using Trizol reagent, according to the manufacturer’s instructions. The amount and purity of total RNA were evaluated with a UV spectrophotometer (NanoDrop 2000, Thermo Fisher Scientific), by A260/280 and 260/230 ratios, considering the cut-off values equal or greater than 2.0 and 1.8, respectively. Extracted RNA samples were treated for 10 min at 37 °C with RNase-free DNase I 0.5 U/μg of RNA. First-strand synthesis of cDNA was performed using a First-Strand cDNA synthesis kit, according to the manufacturer’s instructions (GE Healthcare Life Sciences, Chicago, IL, USA). The quantitative real-time PCR amplification was performed using GoTaq qPCR Master Mix (Promega, Madison, WI, USA) and carried out on a Real-Time PCR System (StepOne; Applied Biosystems, Waltham, MA, USA) under the following conditions: an initial holding stage at 95 °C for 2 min was followed by 40 cycles: denaturation at 95 °C for 15 s, annealing and extension at 60 °C for 1 min. A melt curve was performed (slow ramp of 0.3 °C/20 s to 95 °C). Amplification was performed using primers specific for GAPDH (NM_001289746.1), IL-6 (NM_001318095.1), IL-10 (NM_000572.2), IL-8 (NM 000584.1), FGF2 (NM_001361665.2), OPG (TNFRSF11B: NM_002546.4) and RANKL (TNFSF11: NM_003701.4). Data were analyzed using the StepOne v2.0.2 software (Applied Biosystems). For both control and treated samples, the GAPDH Ct values were subtracted from the genes’ Ct values to obtain the ΔCt value. ΔΔCt values were obtained according to ΔCt (treated samples)—ΔCt (control samples). The relative quantity was calculated according to 2^−ΔΔCt^.

### 4.15. Statistical Analysis

The comparison of the different groups at each experimental time was carried out using a Kruskal–Wallis non-parametric analysis of variance, with Dunn’s post-hoc test, considering an alpha error of 5%. Statistical analysis was performed with the help of GraphPad software Prism 8 (GraphPad Software, Boston, MA, USA).

## 5. Conclusions

Compared to L-PRF, the Alb-PRF protocol produces biphasic membranes with a dense denatured albumin barrier, with a large number of homogeneously distributed nucleated cells. Alb-PRF membranes release lower levels of inflammatory cytokines and higher concentrations of PDGF and can induce the proliferation of human osteoblasts, similar to L-PRF, but with increased stimulation of alkaline phosphatase activity and in vitro mineralization, as well as a positive balance of the OPG/RANKL ratio. These in vitro results suggest that Alb-PRF is a promising material for future clinical studies to determine its suitability for applications involving bone repair.

## Figures and Tables

**Figure 1 ijms-25-11531-f001:**
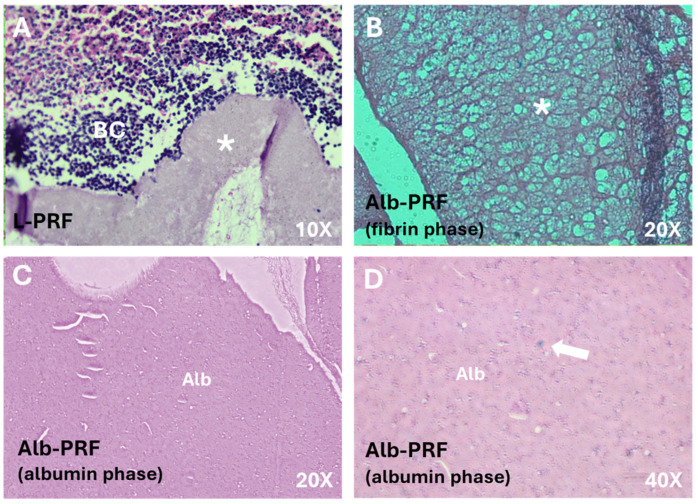
Images of histological sections of the membranes. (**A**) L-PRF membrane at its extremity, where both a dense fibrin network (*) and the buffy coat (BC) can be identified with a high density of cells; Alb-PRF forms a biphasic material with a fibrin network phase shown in (**B**), connected to a dense albumin barrier (Alb) shown in (**C**). This albumin portion is also cellularized (arrow), as it can be seen at a higher magnification (**D**).

**Figure 2 ijms-25-11531-f002:**
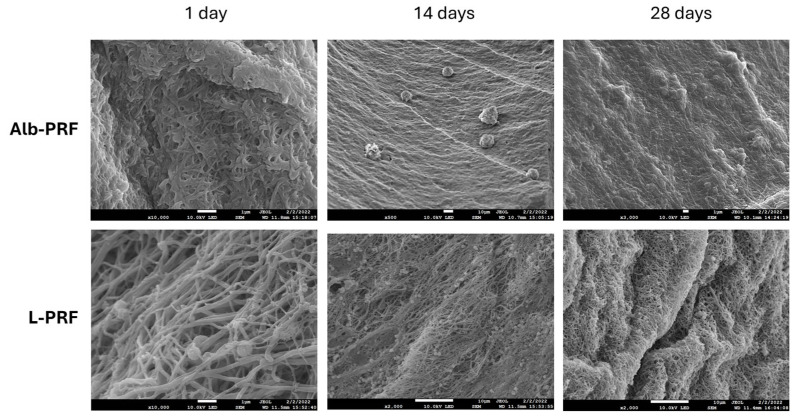
Scanning electron micrographs of the Alb-PRF (**above**) and L-PRF (**below**) membranes after elution in culture medium at times of 1, 14, and 28 days. Images were obtained at 15 kV, at different magnifications (JEOL JSM 7100F, JEOL, Japan). The scale bar in each figure indicates 1 μm.

**Figure 3 ijms-25-11531-f003:**
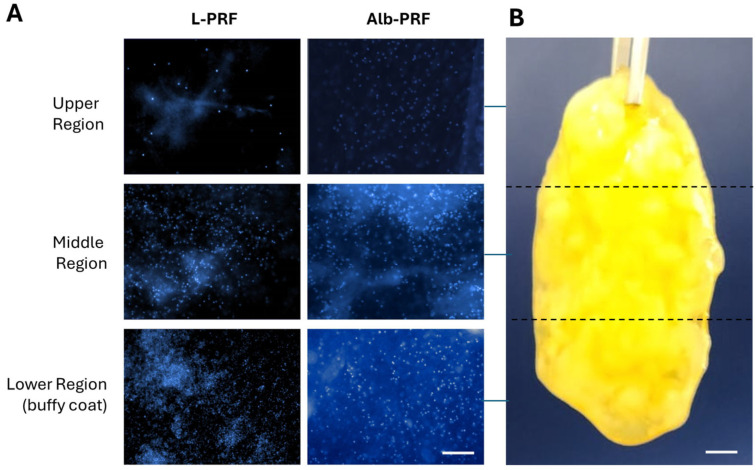
Qualitative panel of cellular distribution in the structure of L-PRF and Alb-PRF membranes (**A**). Images were taken from three different portions of the membranes, as shown for an Alb-PRF membrane with approximately 3 cm of length (**B**). Cell nuclei were evidenced by fluorescence microscopy after staining with DAPI in the upper, middle, and lower regions of the L-PRF membrane and visualization of three random fields of the Alb-PRF membrane (images obtained with a ×20 objective). The scale bar in (**A**) represents 100 μm, while in (**B**) it represents 3 mm.

**Figure 4 ijms-25-11531-f004:**
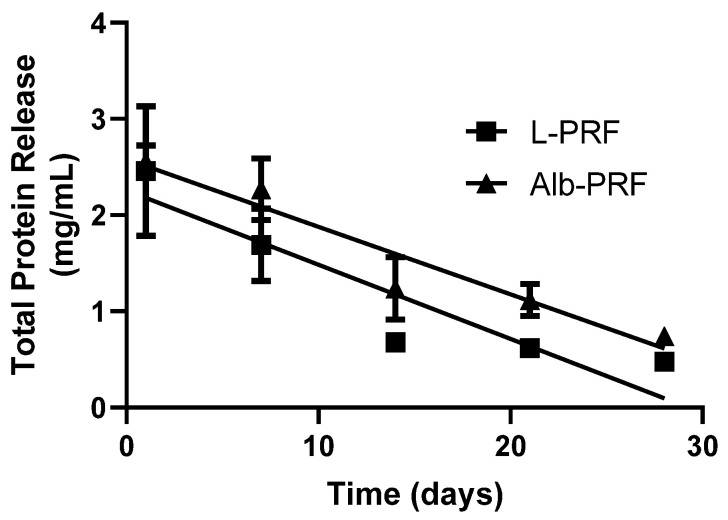
Quantification of total proteins released into the culture medium after exposure to L-PRF and Alb-PRF membranes for 1, 7, 14, 21 and 28 days. Points indicate mean and standard deviation (*n* = 5). The lines present linear regressions with R^2^ values of 0.8293 and 0.8639 for L-PRF and Alb-PRF, respectively.

**Figure 5 ijms-25-11531-f005:**
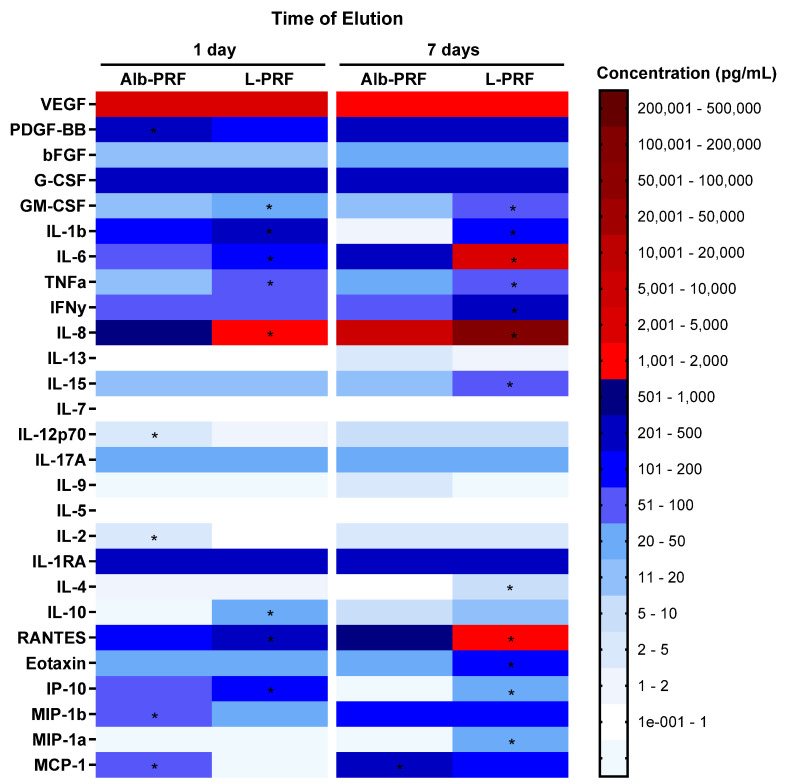
Heatmap of the variation in concentrations of 27 analytes measured in culture media after incubation of L-PRF and Alb-PRF membranes, at times of 1 and 7 days. (*) Significantly different values between these experimental groups (*p* < 0.05).

**Figure 6 ijms-25-11531-f006:**
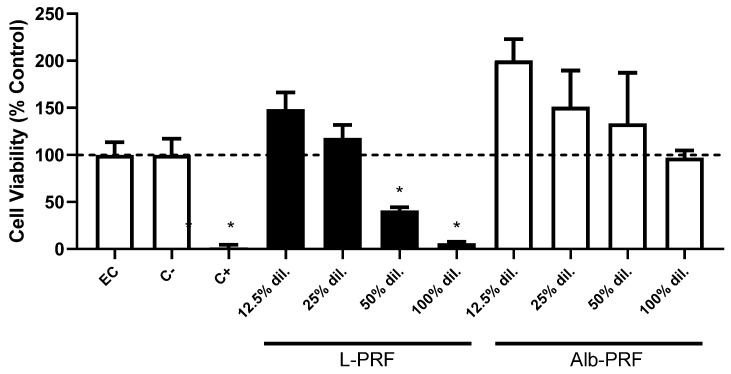
Cell viability of MG-63 exposed to 24 h extracts from L-PRF and Alb-PRF membranes, diluted at different concentrations with culture media, where the percentage dilution indicates the proportion added of the original extraction media (100% dilution). Results are represented as mean and standard deviation (SD) (*n* = 5) of cell viability as a percentage of the unexposed experimental control (EC), as measured by the XTT test. High-density, non-cytotoxic polystyrene beads were employed as negative control (C−), and latex fragments were used as a positive control of cytotoxicity (C+). (*) Significantly different values from all other experimental groups and EC (*p* < 0.05). A dashed line indicates 100% viability (as a percentage of the control group).

**Figure 7 ijms-25-11531-f007:**
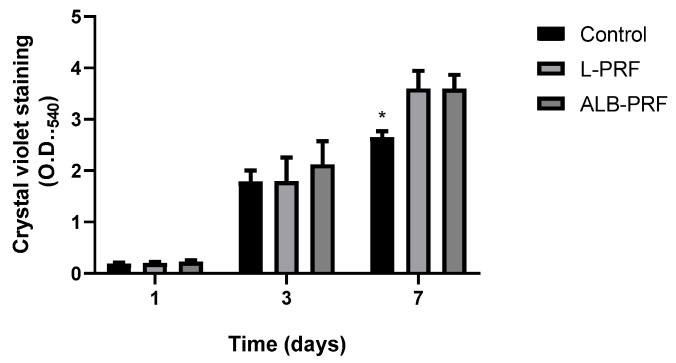
Cellular proliferation of MG-63 osteoblasts exposed to 25% of the 24 h extracts of L-PRF and Alb-PRF at 1, 3, and 7 days, with unexposed cells as a control. Results presented as a mean and SD (*n* = 5) of the staining of DNA by the CVDE assay, which is directly relative to cell density. (*) Significantly different from the other experimental groups, at the same time (*p* < 0.05).

**Figure 8 ijms-25-11531-f008:**
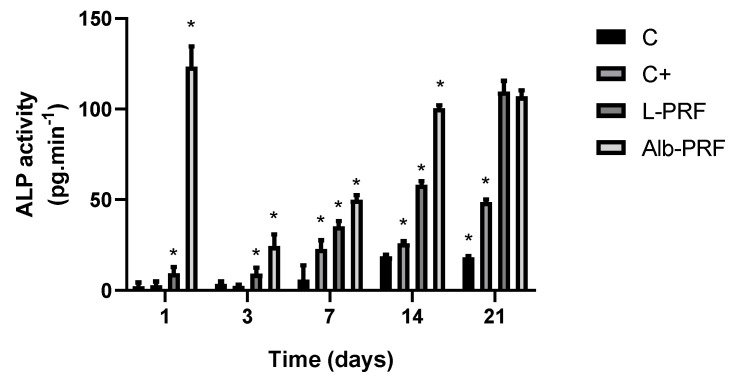
Measurement of alkaline phosphatase concentrations obtained after exposure of MG-63 osteoblasts to 25% conditioned media of each membrane, at 1, 3, 7, 14, and 21 days. Bars indicate mean and standard deviation (*n* = 5). (*) Significantly different values between groups in the same experimental time (*p* < 0.05). Mineralization medium was employed as a positive control (C+).

**Figure 9 ijms-25-11531-f009:**
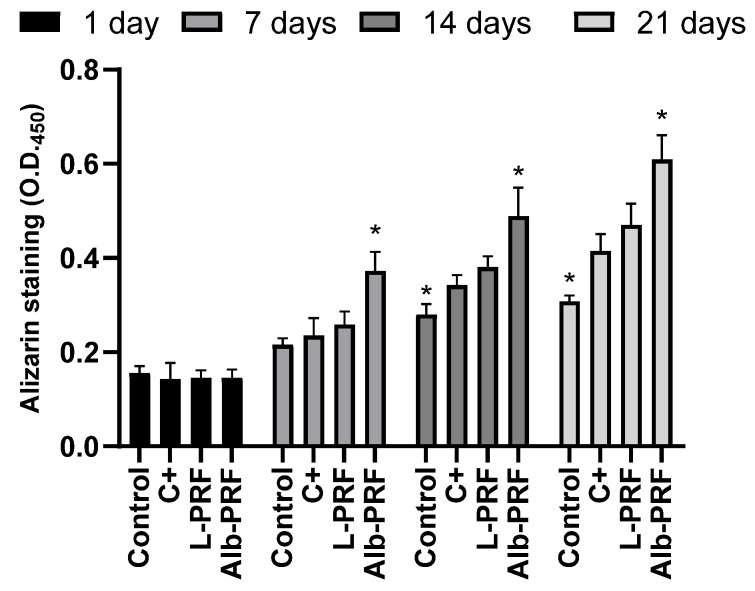
The quantification of mineralization was determined by alizarin red after exposure of MG-63 osteoblasts to 25% dilution of the extracts at 1, 3, and 7 days. The bars indicate the mean and standard deviation (*n* = 3). (*) Significantly different from all other groups at the same experimental time (*p* < 0.05). Mineralization medium was employed as a positive control (C+).

**Figure 10 ijms-25-11531-f010:**
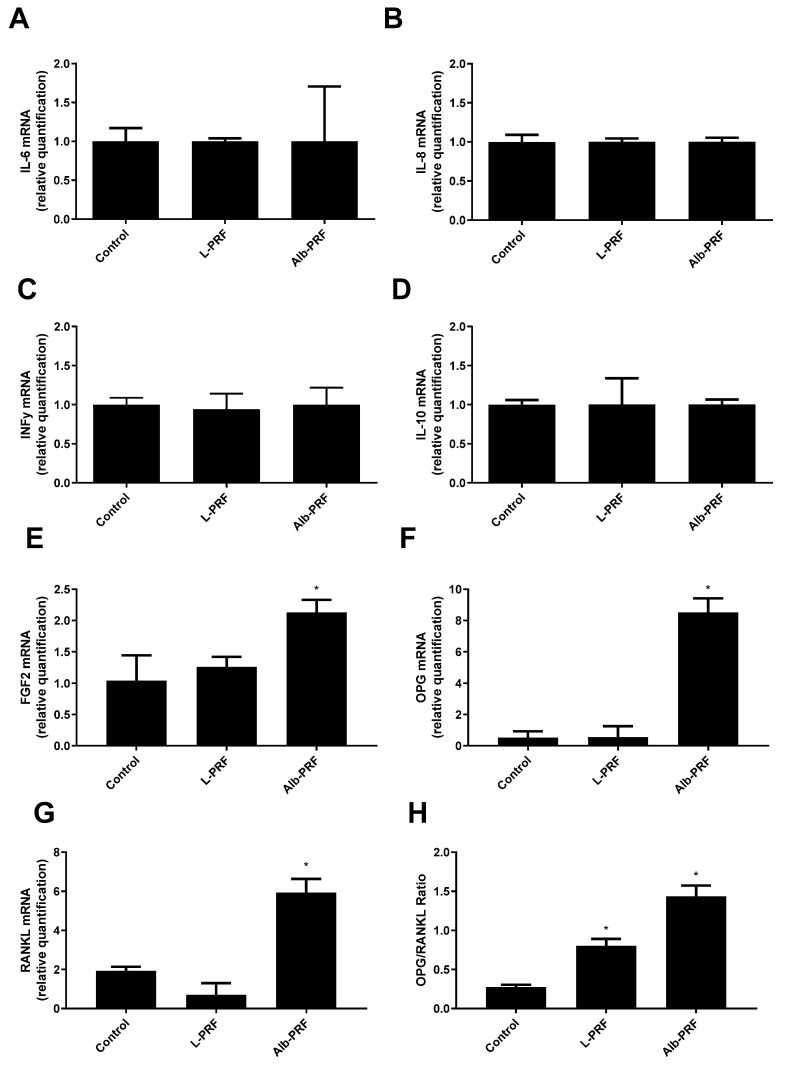
Quantification of mRNA expression of different inflammatory and bone metabolism-related markers, after exposure of MG-63 osteoblasts to 25% conditioned media for 1 day. (**A**) IL-6; (**B**) IL-8; (**C**) IFN-y; (**D**) IL-10; (**E**) FGF2; (**F**) OPG; (**G**) RANKL; the (**H**) OPG/RANKL ratio. Bars indicate the mean and standard deviation (*n* = 3). (*) Significantly different from all other groups (*p* < 0.05). Results were calculated as a proportion of Glyceraldehyde 3-phosphate dehydrogenase (GAPDH) expression.

## Data Availability

Data are available from the corresponding author upon request.

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
