# Peer review of "Investigating the Biological Efficacy of Albumin-Enriched Platelet-Rich Fibrin (Alb-PRF): A Study on Cytokine Dynamics and Osteoblast Behavior"

_ijms, 2024, doi:10.3390/ijms252111531_

Round 1

Reviewer 1 Report

Comments and Suggestions for Authors

This study should be able to interest the broad readers working on bone repairs and could inspire more developments on advanced biomaterials in this field. Given the well-organized structure and sound experiments, I would suggest to publish this paper as it is.

This study improves the weakness of L-PRF by using Alb-PRF for a better mechanical stability, sustained release of bioactive factors, etc and this would facilitate development of it with a broad applications along with biomaterials - ready to be accepted as it is.

The paper is well-structured and consistent with the result provided, given the facts as following:

  1. Histology/SEM/membranes images indicate a better scaffold for bone regeneration.
  2. The level of cytokine and growth factors release was  lowered and increased respectively by Alb-PRF compared to L-PRF, suggesting the potential for the promotion of bone healing and regernation.
  3. The increased of OPG/RANKL by Alb-PRF also shows the potential for an enhanced osteogenesis.

However, it would be better if the author could spend some efforts to improve the quality of this study:

  1. Figure 2, please add scale bar number in the caption - it is not clear in the image
  2. Figure 3B, please add scale bar
  3. Minor suggestion on Figure 6, better to have a line across 100% for easy reading and comparison

Author Response

Comment 1: Figure 2, please add scale bar number in the caption - it is not clear in the image.

Answer 1: Thank you for your observation. We have updated the caption of Figure 2 to include the scale bar information to improve clarity.

Comment 2: Figure 3B, please add scale bar

Answer 2: We appreciate your suggestion. The scale bar has been added to Figure 3B to facilitate better visualization of the cellular distribution.

Comment 3: Minor suggestion on Figure 6, better to have a line across 100% for easy reading and comparison

Answer 3: We agree with your suggestion and have added a line across 100% in Figure 6 to improve readability and ease of comparison.

The authors acknowledge the reviewer for its unbiased review and contribution.

Reviewer 2 Report

Comments and Suggestions for Authors

The manuscript investigates the biological efficacy of albumin-enriched platelet-rich fibrin (Alb-PRF) and its effects on cytokine dynamics and osteoblast behavior. The study compares Alb-PRF with leucocyte and platelet-rich fibrin (L-PRF) in terms of cytokine and growth factor release, as well as osteoblast proliferation, differentiation, and mineralization. The manuscript is clearly written and follows a logical flow, making it easy for readers to follow the methodology and understand the results. This study utilizes various techniques, including histological, scanning electron microscopy, and cytokine release assays, to thoroughly characterize Alb-PRF's structural and functional properties. The results of this project suggest that Alb-PRF holds considerable potential as an improved biomaterial for bone regeneration. Its combination of enhanced structural stability, controlled cytokine release, and superior promotion of osteoblast activity makes it a promising alternative to traditional PRF formulations. The increased OPG/RANKL ratio and improved mineralization capacity further underline its applicability in bone tissue engineering and repair. Future research should focus on validating these findings in vivo, as well as exploring clinical applications of Alb-PRF in various surgical and therapeutic settings.

Q 1: This study does not evaluate the mechanical properties of the Alb-PRF membrane, such as stiffness or elasticity, when interacting with cells. Mechanical properties are crucial for supporting tissue regeneration, particularly in bone, where cells respond to mechanical cues for proper differentiation and mineralization. Please Assess the mechanical properties of the Alb-PRF membrane and discuss how these properties influence cell behavior, particularly in terms of osteogenic differentiation and tissue formation.

Q 2: This study relies on the use of extracts from Alb-PRF for cell studies rather than directly applying the membrane to the cells. Using extracts may not fully replicate the in vivo interaction between cells and the Alb-PRF membrane. Important factors such as the physical interaction between cells and the membrane structure, the localized release of growth factors, and the mechanical properties of the scaffold are not assessed. These factors are critical for tissue regeneration in clinical applications. This study lacks insight into how cells behave in a three-dimensional (3D) environment when interacting with Alb-PRF membranes. Studies where cells are cultured directly on Alb-PRF membranes should be provided.

Q 3: The significantly high alkaline phosphatase (ALP) concentrations observed in the Alb-PRF group after just 1 day of exposure to 25% conditioned media could point to an accelerated osteogenic differentiation process induced by Alb-PRF. While this is an interesting and potentially beneficial outcome, it also raises several points that warrant careful consideration and further discussion.

Future studies should track ALP activity over extended periods to determine whether the early peak in ALP activity is sustained and translates into enhanced bone formation in long-term settings. In vivo studies are crucial for verifying the clinical relevance of this rapid ALP induction.

Comments on the Quality of English Language

Minor revision.

Author Response

Comment 1: This study does not evaluate the mechanical properties of the Alb-PRF membrane, such as stiffness or elasticity, when interacting with cells. Mechanical properties are crucial for supporting tissue regeneration, particularly in bone, where cells respond to mechanical cues for proper differentiation and mineralization. Please Assess the mechanical properties of the Alb-PRF membrane and discuss how these properties influence cell behavior, particularly in terms of osteogenic differentiation and tissue formation.

Answer 1: We acknowledge the importance of mechanical properties in tissue regeneration. The authors agree that the mechanical aspects of Alb-PRF, such as stiffness and elasticity, are crucial for understanding its full potential in supporting bone regeneration. Nevertheless, this specific study focuses primarily on its biological efficacy over osteoblast behavior. After the important suggestion by the reviewer, we plan to evaluate these mechanical properties in future studies, using rheometry to determine how mechanical cues from Alb-PRF influence osteogenic differentiation. A paragraph addressing this topic has been added to the discussion section.

Comment 2: This study relies on the use of extracts from Alb-PRF for cell studies rather than directly applying the membrane to the cells. Using extracts may not fully replicate the in vivo interaction between cells and the Alb-PRF membrane. Important factors such as the physical interaction between cells and the membrane structure, the localized release of growth factors, and the mechanical properties of the scaffold are not assessed. These factors are critical for tissue regeneration in clinical applications. This study lacks insight into how cells behave in a three-dimensional (3D) environment when interacting with Alb-PRF membranes. Studies where cells are cultured directly on Alb-PRF membranes should be provided.

Answer 2: Thank you for your insightful comments regarding the importance of studying cell behavior in direct contact with the Alb-PRF membrane. While we fully agree that such an approach is critical for evaluating cell migration into the scaffold (as in the case of L-PRF) or the barrier effect (as with Alb-PRF), we chose to use indirect exposure to specifically isolate and evaluate other key parameters.

Direct contact may introduce challenges in isolating factors like proliferation, mineralization quantification, and molecule release (e.g., ALP), as these molecules could become entrapped within the fibrin or albumin matrix, affecting accurate measurement. As noted in a recent review (Barbosa et al., JFB 2023:14(10):503), indirect exposure methods are widely used in cell/PRF studies because they allow for the controlled analysis of these parameters without interference from the physical matrix, which could entrap molecules and obscure results. Furthermore, the extraction of mRNA for analysis is more feasible when cells are exposed to conditioned media rather than being embedded in the membrane. Thus, we believe that our methodology provides valuable insights into the effects of Alb-PRF on osteoblast behavior.

Therefore, while we believe that our chosen methodology offers robust and relevant results for these specific parameters, we acknowledge the great value of direct contact assays, as you suggested, which we fully intend to incorporate in future works, including also 3d osteoblast culture, to further enhance the understanding of Alb-PRF's interaction with cells in a three-dimensional environment. We point out this limitation in the revised version of the manuscript.

Comment 3: The significantly high alkaline phosphatase (ALP) concentrations observed in the Alb-PRF group after just 1 day of exposure to 25% conditioned media could point to an accelerated osteogenic differentiation process induced by Alb-PRF. While this is an interesting and potentially beneficial outcome, it also raises several points that warrant careful consideration and further discussion. Future studies should track ALP activity over extended periods to determine whether the early peak in ALP activity is sustained and translates into enhanced bone formation in long-term settings. In vivo studies are crucial for verifying the clinical relevance of this rapid ALP induction.

Answer 3: This is an interesting observation. As we preserved the conditioned media from the mineralization assays (presented in Figure 9) at -80°C, we were able to assess the alkaline phosphatase activity from exposed cells also at 14 and 21 days, which are very relevant times of in vitro mineralization of osteoblasts according to several different studies. These data are presented in the revised Figure 8.

The authors thank the reviewer for the unbiased evaluation and the important observations that improved the scientific quality of the revised manuscript.

Round 2

Reviewer 2 Report

Comments and Suggestions for Authors

The questions are discussed in the revised paper, and added some limitations. 

Comments on the Quality of English Language

Minor revision needed.